# *Carnivore protoparvovirus 1* (CPV-2 and FPV) Circulating in Wild Carnivores and in Puppies Illegally Imported into North-Eastern Italy

**DOI:** 10.3390/v14122612

**Published:** 2022-11-23

**Authors:** Stefania Leopardi, Adelaide Milani, Monia Cocchi, Marco Bregoli, Alessia Schivo, Sofia Leardini, Francesca Festa, Ambra Pastori, Gabrita de Zan, Federica Gobbo, Maria Serena Beato, Manlio Palei, Alessandro Bremini, Marie-Christin Rossmann, Paolo Zucca, Isabella Monne, Paola De Benedictis

**Affiliations:** 1National Reference Centre/WOAH Collaborating Centre for Diseases at the Animal-Human Interface, Istituto Zooprofilattico Sperimentale Delle Venezie, 35020 Legnaro, Italy; 2Istituto Zooprofilattico Sperimentale Delle Venezie, Sezione Territoriale di Udine, 33030 Basaldella di Campoformido, Italy; 3Central Directorate for Health, Social Policies and Disabilities, Friuli Venezia Giulia Region, 34123 Trieste, Italy; 4Biocrime Veterinary Medical Intelligence Centre, c/o International Police and Custom Cooperation Centre, Thörl-Maglern, 9602 Arnoldstein, Austria; 5Agiculture, Forestry, Rural Areas Veterinary Department, Land Carinthia, 9020 Klagenfurt, Austria

**Keywords:** CPV-2, FPV, illegal trade, companion animals, wildlife, spillover

## Abstract

The illegal trade of animals poses several health issues to the global community, among which are the underestimated risk for spillover infection and the potential for an epizootic in both wildlife and domestic naïve populations. We herein describe the genetic and antigenic characterization of viruses of the specie *Carnivore protoparvovirus 1* detected at high prevalence in puppies illegally introduced in North Eastern Italy and compared them with those circulating in wild carnivores from the same area. We found evidence of a wide diversity of canine parvoviruses (CPV-2) belonging to different antigenic types in illegally imported pups. In wildlife, we found a high circulation of feline parvovirus (FPV) in golden jackals and badgers, whereas CPV-2 was observed in one wolf only. Although supporting a possible spillover event, the low representation of wolf samples in the present study prevented us from inferring the origin, prevalence and viral diversity of the viruses circulating in this species. Therefore, we suggest performing more thorough investigations before excluding endemic CPV-2 circulation in this species.

## 1. Introduction

Parvoviruses are non-enveloped viruses with a short genome of non-segmented single-stranded DNA encoding for two open reading frames (ORFs), among which ORF1 encodes for non-structural proteins NS1 and NS2 and ORF2 encodes for the capsid proteins VP1 and VP2 [1]. Parvoviruses of companion animals belong to the species *Carnivore protoparvovirus 1*, genus *Protoparvovirus*, and family *Parvoviridae*. Canine parvovirus type-2 (CPV-2) arose in the mid-1970s as a variant of a virus similar to but distinct from FPV [2]. Historically, typing of CPV-2 was based on the antigenic variability of strains assessed using monoclonal antibodies or amino acid (aa) substitutions (at aa residues 297 and 426) within the gene coding for the major capsid protein VP2 [3], although phylogenetic analyses based on either the VP2 or the whole genome do not completely reflect the antigenic properties [1]. Since its first emergence, CPV-2 was selected in dogs giving rise to several antigenic and genetic variants: CPV-2a, CPV-b, and the most recent subtypes new CPV-2a, new CPV-2b and CPV-2c, that gradually replaced the original type [3,4]. Nowadays, CPV-2 is occasionally detected worldwide likely due to its use as attenuated live vaccine [5,6]. Compared to the original CPV-2, the new variants recovered the ability to infect felids and showed increased pathogenicity [2,7].

Regardless of the classification used and based on the sequences publicly available, parvoviruses fail to show any significant geographical clustering in dogs, thus suggesting high admixing between populations due to the extensive movement of persons and their pets and, very likely, to the ever-growing commercialization, legal and not, of pups [1]. In this context, it hast been suggested that the black market of puppies might contribute to the spread and evolution of CPV-2 in European count ries with high vaccination coverage, such as Italy [6].

The European Commission considers the illegal trade of pets an emerging risk for Member States. Indeed, considering the sanitary impacts, the illegal distribution not only poses risks to the pet itself due to the possible bad practices in keeping, breeding and transporting, but also represents a risk for the introduction of epizootic and zoonotic diseases into free areas [8,9]. Within this frame, the important sanitary impacts driven by the illegal transport of puppies through Northeastern Italy has been underlined. Indeed, a three-year survey demonstrated that illegally imported puppies displayed poor vaccine immunity—with canine parvovirus and giardia recognized as the infections most frequently associated to fatal gastroenteritis and, most importantly, identified *Salmonella* and *Microsporum canis* as major zoonotic pathogens [10]. In 2015, the cat and dog trade involved 61 million dogs and 67 million cats in twelve EU Member States, representing € 1.3 billion and generating a direct employment of about 300,000 workers [8]. The illegal trade of puppies also represents a source of illegal market, thus an unfair competition for complaint breeders and sellers [11].

Both CPV-2 and FPV have been detected in wild carnivores of different genera across the world, with cross-species transmission at the domestic-wildlife interface still evident in some countries, such as South America, making virus dynamics and evolution rather complex [1]. In other areas, including most European countries investigated so far, CPV-2 and FPV strains have mostly become endemic in wild reservoirs, even if sporadic spillover events are still detected [12]. As for dogs and cats, the pathogenicity of parvoviruses in wildlife is variable, spanning from asymptomatic infection to severe diarrhea in pups, possibly affecting the health of fragile populations [12,13,14].

In the present study, we characterized viruses of the specie *Carnivore protoparvovirus 1* (FPV and CPV-2) circulating in wild carnivores from Friuli Venezia-Giulia (North Eastern Italy) and compared them to the strains that have been introduced in the same area through the illegal trade of puppies [10].

## 2. Materials and Methods

The sample set included (i) n = 256 feces samples collected from puppies admitted in quarantine facilities in Friuli Venezia-Giulia between 2018 and 2021 after their illegal introduction from Central and Eastern Europe and (ii) n = 79 intestines collected post-mortem in case of the animal death during the observation period [10]. In addition, we analyzed intestinal samples of wild carnivores collected since 2021 in the framework of passive surveillance for rabies from the same region. These include n = 55 from Eurasian badgers (*Meles meles*), n = 21 from red foxes (*Vulpes vulpes*), n = 12 from golden jackals (*Canis aureus*), n = 5 from gray wolves (*Canis lupus lupus*), and n = 4 from beech martens (*Martes foina)*. Samples were homogenized and nucleic acids were extracted using QIAsymphony DSP Virus/Pathogen Midi kit on the QIAsymphony SP instrument (QIAGEN, Hilden, Germany) or MagMAX Viral/Pathogen II on KingFisher Magnetic Particle Processors (Thermo Fisher Scientific, Waltham, Massachusetts) for dog and wildlife samples, respectively. All samples were screened using molecular testing for canine parvovirus and feline panleukopenia virus [6] using the QuantiFast^®^ Pathoghen PCR+IC (QIAGEN, Hilden, Germany) as amplification kit and CFX 96 BIO-RAD (BIO-RAD, Hercules, CA, USA) as platform.

We characterized the near complete genome of all wildlife strains and of strains from selected samples of seventy-four positive dogs. More specifically, samples from dogs were selected using a random stratified sampling, according to the year of detection and the type of sample. In order to do this, we used a target PCR approach and Next Generation Sequencing (NGS). Primers were modified from Perez et al. (2014) [15] or designed de novo to obtain two or three overlapping amplicons using alternative protocols (Table 1 and Table 2). 

All PCR protocols were run in a final volume of 25 µL, using 1 to 5 ng of sample DNA, 0.7 µM of each primer, 1X PCR buffer, 0.8 M MgCl_2_ and 1 U of Platinum Taq polymerase (Invitrogen). The amplification included 5 min at 94 °C, followed by 40 cycles at 94 °C for 30 s, 58 °C for 30 s and 72 °C for 3 min and by a final extension of 10 min at 72 °C. For sequencing, we pooled amplicons belonging to the same sample in equimolar ratio, prepared libraries using the Nextera XT DNA sample preparation kit (Illumina, San Diego, CA, USA) and processed them on an Illumina MiSeq platform with the MiSeq reagent kit V3 (2 × 300) or V2 (2 × 250) (paired-end [PE] mode; Illumina, San Diego, CA, USA) following the company’s instructions. 

After assessing the quality of raw reads with FastQC v0.11.7 (https://www.bioinformatics.babraham.ac.uk/projects/fastqc/, accessed on 23 September 2022), we used scythe v0.991 (https://github.com/vsbuffalo/scythe, accessed on 23 September 2022) to clip them from Illumina Nextera XT adaptors sequences (Illumina, San Diego, CA, USA) and cutadapt v2.10 to trim the adaptors and filter raw reads with length below 80 nucleotides and Q score below 30. We then generated complete genomes through a reference-based approach using BWA v0.7.12 (https://github.com/lh3/bwa, accessed on 23 September 2022) [16]. Finally, we used Picard-tools v2.1.0 (http://picard.sourceforge.net (accessed on 23 September 2022)) and GATK v3.5 (https://github.com/moka-guys/gatk_v3.5, accessed on 23 September 2022) to process alignments, loFreq v2.1.2 (https://github.com/CSB5/lofreq, accessed on 23 September 2022) to call Single Nucleotide Polymorphisms (SNPs) and an in-house script to obtain consensus sequences, setting 50% of allele frequency as threshold for base calling and 10X as the minimum coverage.

In order to investigate the diversity of parvovirus strains circulating in the area, we performed genetic and phylogenetic analyses for the whole genome and the complete VP2 gene, which is more widely used across the literature. Datasets included positive original samples and the first three non-identical best matches for each sequence, as determined using BLAST reference strains that were previously associated either with dogs in the study area [6] or with wildlife across the world. The selection from the public database included sequences from Italy or wild species tested positive in our study. Sequences were aligned using the G-INS-1 parameters implemented in Mafft [17] and Maximum likelihood (ML) nucleotide phylogenetic trees were inferred using PhyML (version 3.0), employing the GTR+Γ4 substitution model, a heuristic SPR branch-swapping algorithm and 1000 bootstrap replicates [18]; the obtained trees were then graphically edited using iTOL [19]. In order to achieve the typing of the CPV-2/FPV strains, we derived amino acidic sequences using MEGA6 and considered VP2 amino acid residues at positions 87, 297, 300, 305, 426 and 555, as described elsewhere [1].

## 3. Results

Overall, we screened n = 387 fecal samples and intestines from dogs and different wildlife species, achieving a total of n = 298 positive samples among dogs (n = 290/343, 84.5%), Eurasian badgers (n = 4/55, 7.2%), golden jackals (n = 3/12, 25%), and grey wolves (n = 1/5, 20%). All samples from red foxes (n = 21) and beech martens (n = 4) tested negative (Table 3). Pathological findings in positive dogs were in accordance with a moderate to severe hemorrhagic/necrotic-hemorrhagic enteritis. Similarly, the positive wolf was an adult male presenting a severe hemorrhagic enteritis, whereas the golden jackals were all young individuals and did not show any gut lesion. On the other hand, we could not evaluate the association between virus positivity and clinical or pathological conditions of badgers, which were submitted for rabies surveillance as carcasses at various degrees of putrification, or after severe car accidents, both conditions preventing reliable necropsies.

Seventy-four positive samples from dogs and all positive samples from wildlife were further characterized throughout whole genome sequencing. The majority of CPV-2 detected and characterized in dogs were new CPV-2a (Table 3) followed by CPV-2, new CPV-2b, CPV-2b, and CPV-2c. Whereas viral strains from wildlife were characterized as FPV (from Eurasian badgers and golden jackals) and CPV-2c (from one grey wolf) (Table 3; Figure 1).

All positive samples were sequenced using one of the three molecular approaches described, obtaining consensus sequences of a total length of 4000–4300 nucleotides that mostly excluded only non-coding terminal regions (160–400 bp on 5′ and 140 bp on 3′). However, we failed to amplify around 2000 bp on 5′ in nine samples, including four dogs, the wolf, one jackal and three badgers, and on 3′ in three badger samples. In addition, another jackal’s sample did not provide an interpretable sequence across 1000 bp towards the 5′ end. The obtained sequences have all been deposited under the GenBank accession numbers OP587964 to OP588036 and OP595737 to OP595745. Alignments used for phylogenetic analyses accounted for 130 and 230 sequences for the whole genome and the VP2 region, respectively. Of these, 73 and nine, respectively, have been obtained in the present Investigation. The topology of the phylogenetic trees based on the whole genome sequences and on the VP2 gene sequences were comparable, showing that sequences of viral strains from wildlife were included in a separate branch, along with the sequences of other FPV strains, except for the one from the grey wolf, included in a cluster within the sequences of CPV-2 strains. Sequences related to illegally imported puppies displayed a high differentiation of canine parvoviruses included in several clusters across the whole phylogenetic tree (Figure 1 and Figure 2). 

Almost all published available CPV-2/FPV sequences detected in the Italian domestic dogs and cats, as well as in European wildlife, accounted for partial genomes. In this framework, phylogeny based on the VP2 gene sequences provided better reference to interpret phylogenetic trees. Indeed, FPV sequences from Eurasian badgers generated in this study formed a stand-alone cluster within FPV considering the whole genome (Figure 2). Conversely, considering the VP2 only, the FPV strains sequenced from wildlife appeared scattered in the phylogenetic tree, showing high correlation with strains described in Italian cats but also in Eurasian badgers and stone martens from Spain and the UK (Figure 1). Interestingly, two of these sequences (OP595737 and OP595738) were identical to Italian FPVs associated with cats at the VP2 level (KX434461 and KX434462, respectively). 

FPV sequences generated from golden jackals (n = 3) were not related to each other (Figure 1). One (OP595745) clustered with the FPV sequence (OP595740) from Eurasian badger characterized in this study and with the FPV sequences from Italian cats and wildlife, a second one (OP588006) with the FPV sequence from Canadian wildlife, and the last one (OP587998) with the FPV sequences from Italian domestic cats. As for FPV strains from Eurasian badgers, this last sequence (OP587998) was highly related with sequences obtained from cats only at the level of VP2, while it occupied a basal branch with no evident clustering upon analyses using the complete genomes (Figure 2).

Similarly, the analysis of the VP2 sequences allowed us to show that most clusters identified in dogs in our study were highly related with strains already described in Friuli Venezia-Giulia or, more generally, in Italy (Figure 1). Interestingly and similarly to what previously noticed by [6], in the present study the VP2 phylogenetic tree showed that the CPV-2 strains sequenced from illegally imported dogs are grouped according to the antigenic type (Figure 1). Considering the antigenic characterization, 55 sequences classified as new CPV-2a, six as CPV-2b, two as new CPV-2b, and two as CPV-2c (Figure 1). The high prevalence of the new CPV-2a variant in illegally imported dogs is similar to the one recently observed in dogs in the same Italian geographical area [6]. Of note, nine sequences obtained from puppies clustered within CPV-2 strains actively circulating in dogs in the 1980s and were included in widely used attenuated vaccines (Figure 1 and Figure 2). By analyzing their amino acid sequence, we were able to confirm their antigenic classification as CPV-2 strains (Figure 1). Eight out of nine sequences, clustering with previously characterized CPV-2 in Italy, showed mutations I219V and Q386K typical of one of the live-attenuated vaccines available in the market (MG264079) [20]. Interestingly, four of the identified CPV-2 strains were associated with severe clinical signs and mortality in puppies, three of them (including two sequences displaying the two vaccine signatures as of MG264079) in absence of other pathogens detected (data not shown). 

Finally, the VP2 phylogenetic tree evidenced that CPV new-2a strains identified in the present study from illegally imported dogs make up distinct groups reflecting the detection year to a certain extent (Figure 1). Interestingly, the 2018 new CPV-2a cluster groups together with two new CPV-2a strains of the same year and same region (Friuli Venezia Giulia) that were reported to be suspected of illegal importation from East Europe most probably [6]. 

The strain detected in the grey wolf was the only CPV-2 we found in the screened widlife (OP595742). This sequence was antigenically classified as CPV-2c and phylogenetically closely related with Asian-origin CPV-2c circulating in dogs, showing a maximum nucleotide identity with dog strains of 99.9% across the whole genome and in the VP2 region (Figure 1 and Figure 2). Analyses of the VP2 highlighted that a related sequence (MT454909) had already been associated with wolves in Southern Italy in 2020 (Figure 1) [13]. The analysis of aa residues showed that both the OP595742 sequence identified in the present study and the MT454909 sequence [13] retained tyrosine and threonine at positions 324 and 440 of VP2, respectively, contrasting with the puppies’ sequences retaining only threonine at position 440, out of the three signatures likely caused from vaccine immune pressure (F267Y/Y324I/T440A) [21]. 

## 4. Discussion

This study provides a snapshot of parvoviruses circulating in wild carnivores from Friuli Venezia-Giulia, in comparison with strains imported in the region from Central and Eastern Europe through the illegal market of pups. Despite the vaccination of puppies against canine parvovirus is widely implemented in North Eastern Italy, this practice likely represents a major evolutionary source of old and new strains [6]. The high prevalence of CPV-2 in the tested puppies highlights the risk for domestic dogs posed by the illegal introduction. Indeed, genetic and phylogenetic evaluations of the samples sequenced from illegally traded dogs confirmed that the CPV-2 strains introduced in the area of study belong to several groups across the phylogenetic tree of CPV-2, with eight to twelve clusters identified using the complete VP2 and the whole genome, respectively. Among these, most samples were included in three clusters identified almost exclusively in Italy, with the exception of a single sequence found in Hungary in 2008. Nine sequences clustered with original strains CPV-2 circulating in the 1980s [1], even if most of them formed a sister clade that was exclusive of this work. Similar sequences were reported from neighboring areas in Italy, in particular in the Veneto region and were classified as being likely vaccine strains [6]. Indeed, viruses included in several live attenuated vaccines (shown with red checkmarks in Figure 1) cluster within the original clade of CPV-2. These vaccines are still widely used worldwide, because they induce a strong long-lasting immunity without inducing severe clinical signs [22]. However, the virus replicates within the host so that shedding is frequent after inoculation [23,24]. Because CPV-2 strains do not seem to be actively circulating in European dogs, it is likely that our findings also relate to post-vaccination shedding, as elsewhere already assumed [6]. In this context, it is noteworthy that eight out of nine typed sequences presented two mutations that are included in the patent of a widely used CPV-2 vaccine (MG264079) [20], strongly supporting our hypothesis. Of note, although vaccine replication and shedding are considered non-pathogenic for puppies, we found three cases indicating severe signs of gastroenteritis and mortality linked with vaccine virus. This finding is only partially in line with previous field detections, linking the shedding of vaccine strains with mild diarrhea in pups [25]. In this context, we might not exclude that clinical signs in our study were exacerbated by the poor health of pups under investigation, mostly younger than the recommended vaccination age and transported illegally in poor conditions and not in compliance with animal welfare and sanitary requirements [10].

Despite the whole genome provided a much better resolution of the phylogenetic relationship between viruses identified in the area, the phylogenetic tree based on the VP2 allowed a clearer interpretation thanks to the presence of a greater number of sequences from other areas, especially Italian and European from both dogs and wildlife. Indeed, there are still few studies that provide the complete genome of *Carnivore protoparvovirus 1*, despite they are progressively increasing. In this context, our sequencing protocols allowed us to characterize successfully most strains and were able to provide a cost-effective tool for future studies as well. 

The analyzed CPV-2 strains were associated with most antigenic types currently circulating worldwide (CPV-2, new CPV-2a, CPV-2b, new CPV-2b and CPV-2c), among which new CPV-2a was the most widespread in the area, as previously reported in the same geographic area (North East Italy) [6]. 

In addition, our data support that this subtype completely replaced CPV-2a in the area, while CPV-2b still co-circulates with the recently emerged subtype new CPV-2b. The differentiation of antigenic types mirrored only partially the topology of the phylogenetic tree [1]. In particular, all CPV-2 strains clustered with the “original” clade and all sequences classified as CPV-2c were included within a single cluster diverging from the others. CPV2-b also clustered together, with strains of the new diverging subtype also in the phylogenetic tree. In 2008, CPV-2b reappeared in Italy after ten years, and since then it has maintained a low prevalence [6,26], raising several doubts on the origin of such a renewed circulation. Indeed, CPV-2b has been introduced as vaccine component, due to its notable capability of inducing a broad-spectrum immunity against both CPV-2a and CPV-2c strains [27]. The origin of the CPV-2b sequences identified in the illegally imported puppies in the present study could therefore be due to vaccine strain occurrence rather than to an active circulation of the “old” CPV-2b variant. Of note, all dogs carrying CPV-2b sequences displayed no signs of acute gastroenteritis. Finally, clusters of sequences typed as CPV-2a were sparse across the phylogenetic tree, although clustering together according to the detection year (Figure 1).

Overall, our phylogenetic and genetic data support the assumption that strains detected and sequenced in imported dogs are highly related to what previously observed in the local canine population, suggesting that either the illegal pet trade accounts for most of the variability in parvoviruses in the area, or else that similar strains are circulating in Central and Eastern Europe. However, more recent data from these areas are needed to corroborate one of the two hypotheses. 

In order to test the assumption that parvoviruses introduced in the area with the trade of puppies might pose a risk for local wildlife, we analyzed a large repository of intestinal samples collected from diverse carnivore species widespread in the region. We found only one sample positive for CPV-2 in a grey wolf out of five analyzed in 2022, belonging to the antigenic type CPV-2c. Despite this type was the least represented in dogs from our investigation, this strain was correlated with sequences detected in Asia and also reported in Italian domestic dogs. Similarly to what found elsewhere in Southern Italy, the CPV-2c detected in the wolf retained two aa residues out of the three that had probably emerged from vaccine immune pressure, and may represent current circulation in wildlife following an ancestral spillover event [13,21]. However, investigating the ecopathology of this virus in the grey wolf and its relationship with the domestic dogs deserves a wider investigation. 

No other species was found positive for the presence of CPV-2, differently to that described elsewhere [13,14]. On the other hand, we found that FPV circulates among carnivores in the study area, with percentage of positivity of 7.2 detected in Eurasian badgers, consistently with what reported in Spain [14]. Similarly, a higher circulation of FPV compared to CPV has been described in Portugal, together with a high frequency in Eurasian badgers [28]. To the best of our knowledge, this is the first report of golden jackals resulted positive for FPV in Europe, with the highest percentage of positivity corresponding to 25% (3/12), compared to the other tested carnivores. The high diversity of FPV found in jackals could be explained either by the repopulation of the areas with individuals from different populations or by multiple cross-species transmission. Remarkably, several evidences suggest interactions between jackals and domestic animals, including cats. In addition, the scavenger-like feeding behavior performed by golden jackals in anthropogenic territory may provide the opportunity for spillover from domestic animals to wild carnivores and vice versa. Of note, all foxes tested negative in our study despite the number of individuals tested, thus excluding a prevalence of at least 15% considering the estimated fox population in Friuli Venezia Giulia. This result was expected based on previous findings estimating 2.8% prevalence of FPV in red foxes in other areas of Italy [13] and might be due to the fox susceptibility to *Carnivore protoparvovirus 4* species, of which the prototype virus has been identified in a red fox [29]. Nevertheless, enhanced surveillance in the red fox should be implemented in light of its susceptibility to both FPV and CPV-2 viruses [14,28] and its well-known proximity with human settlements.

Interestingly, FPV sequences detected in this study did not all cluster together based on the widely used VP2, but were correlated with viruses found in both wildlife and domestic cats in Italy and abroad (Figure 1). Similarly, no sequences shared 100% ID based on the whole genome, which suggests a considerable mixing of strains within wildlife populations, including possible cross-species transmissions in the wild and across the interface with domestic cats, as already suggested [14]. Finally, all FPV shared the classical antigenic signature described in domestic animals across all the amino acid residues analyzed.

## 5. Conclusions

In this study, we showed that puppies illegally imported in North-Eastern Italy introduce a wide diversity of canine parvoviruses, belonging to different antigenic types, among which the new CPV-2a is the most represented. However, this diversity matches the one reported elsewhere in Italy, preventing us further speculations.

In wildlife, we found evidence for the infection with CPV-2 only in one wolf, showing a high correlation with a cluster already described in the same species in Southern Italy. Severe clinical signs and mortality associated with original and vaccine strains might be explained by the young age and poor health conditions of the illegally traded pups. Negative findings in foxes and Eurasian badgers suggested that no CPV is circulating at high prevalence in these species in Friuli Venezia Giulia. On the other hand, we found a significant circulation of FPV in golden jackals and, to a lesser extent, in badgers. As for CPV-2, FPV found in wildlife clustered with sequences from wild as well as domestic carnivores, thus describing a complex ecology of *Carnivore protoparvovirus 1*. This ecological complexity is further enhanced by the rapid viral spatial movement associated with the illegal trafficking of puppies from Eastern to Western Europe.

## Figures and Tables

**Figure 1 viruses-14-02612-f001:**
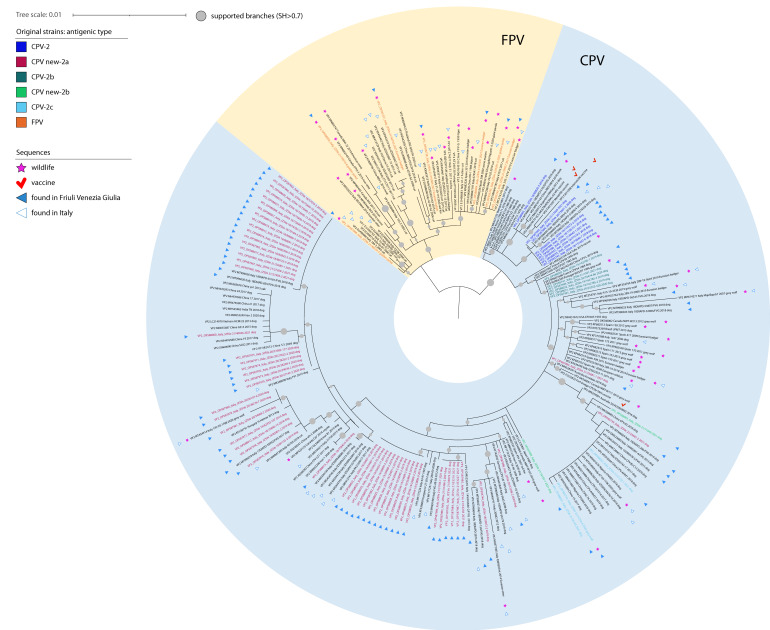
ML phylogenetic tree of *Carnivore Protoparvovirus 1* analyzed in this study, based on the complete VP2 gene sequences, divided in canine (blue) and feline (yellow) parvoviruses. Branches supported with SH value higher than 0.7 are marked with grey circles, with size proportional to the SH value. Original strains sequenced for this work are shown with colored labels based on their antigenic classification as CPV-2 (blue), new CPV-2a (red), CPV-2b (petrol), new CPV-2b (green), CPV-2c (light blue). Sequences (original and reference) associated with wildlife and found in the study area are indicated with pink stars and blue circles, respectively (full for the region of Friuli Venezia-Giulia and empty for some other areas in Italy), while vaccine strains are marked with a red checkmark.

**Figure 2 viruses-14-02612-f002:**
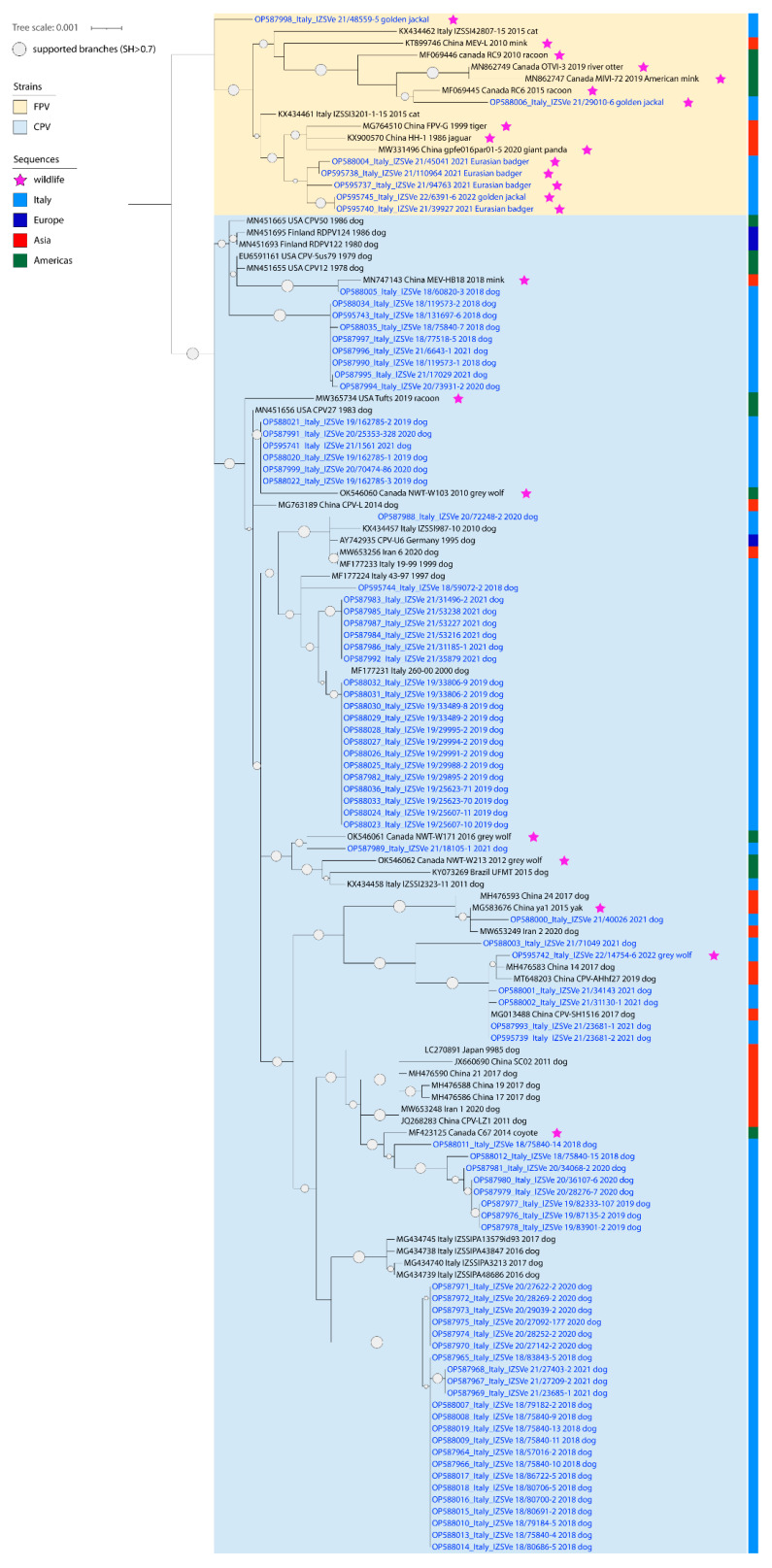
ML phylogenetic tree of *Protoparvovirus 1* analyzed in this study, based on the complete genome, divided in canine (blue) and feline (yellow) parvoviruses. Branches supported with SH value higher than 0.7 are marked with grey circles, with size proportional to the SH value. Original strains sequenced for this work are shown with blue labels. Sequences (original and reference) associated with wildlife and found in the study area are indicated with pink stars.

**Table 1 viruses-14-02612-t001:** Primers used within the study.

Primer	Sequence 5′→3′	Nucleotide Positions	Direction	Reference
NS-Rext	GAAGGGTTAGTTGGTTCTCC	2441–2460	reverse	[15]
F194short	ATAAAAGACAAACCATAGACCGT	194–223	forward	Modified from [15]
NS-Fext	GACCGTTACTGACATTCGCTTC	206–227	forward	
2161For	TTGGCGTTACTCACAAAGACGTGC	2161–2184	forward	
4823Rev	ACCAACCACCCACACCATAACAAC	4800–4823	reverse	
3475R	GTTGGTGTGCCACTAGTTCCAGTA	3452–3475	reverse	
CPV2-2776midF	ATCTTGCMCCAATGAGTGATG	2776–2797	forward	This study
CPV2-4928R	TGGTAAGGTTAGTTCACCTTATA	4905–4928	reverse	

**Table 2 viruses-14-02612-t002:** Protocols used for sequencing the complete genome of *Carnivore protoparvovirus 1*.

Protocol	Primer Combination for 5′ Fragment	Amplicon Size	Primer Combination for 3′ Fragment	Amplicon Size
1	F194short forward + NS-Rext reverse → In case of PCR failure use protocol 2	2400	2161 For forward + 4823 Rev reverse → In case of PCR failure use protocol 3	>2700
2	NS-Fext forward + NS-Rext reverse	2200		
3			2161For forward + 3475 R reverse	1314
			CPV2-2776midF forward + CPV2-4928R reverse	2150

**Table 3 viruses-14-02612-t003:** Details of samples including the genetic and antigenic characterization of positive samples.

Host	N of Positive/Tested Samples	% of Positivity	Viral Type/Variant (% Out of Positive Samples)
Dog	290/343	84.5	CPV-2 (12.2), new CPV-2a (74.3), CPV-2b (8.1), new CPV-2b (2.7), CPV-2c (2.7)
Eurasian badger	4/55	7.2	FPV (100)
Red fox	0/21	0	-
Golden jackal	3/12	25	FPV (100)
Grey wolf	1/5	20	CPV-2c (100)
Beech marten	0/4	0	-

## Data Availability

Sequences have been deposited in GenBank public database under accession numbers OP587964 to OP588036 and OP595737 to OP595745.

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
