# Peer review of "Carnivore protoparvovirus 1 (CPV-2 and FPV) Circulating in Wild Carnivores and in Puppies Illegally Imported into North-Eastern Italy"

_viruses, 2022, doi:10.3390/v14122612_

Round 1

Reviewer 1 Report

The study by Leopardi et al. describes the circulation of canine parvoviruses and feline parvovirus in different animal species. The authors have also tried to evaluate how illegal import of animals can be source of different viral strains carrying novel antigens. The study is indeed important, however, I have some queries: 

In the introduction section, the authors have given the details of structure, prevalence and classification of CPV and FPV. The authors have highlighted the illegal import of animals in the title but have not given its description in the introduction section. To make it more interesting to the readers and to highlight the consequences of illegal import, the authors should also include a paragraph highlighting the status, consequences and associated economic losses due to illegal import of animals. 

In Results section:

The percent positivity of red foxes was found to be zero. However, this point has not been discussed much by the authors. Please include some more discussion by highlighting the possible reasons like lack of receptors etc.

Some of the data in discussion has been explained upto a level that it seems to be the repetition of results. Please get rid of that part. The authors should put more stress on discussion part rather than explaining the results.

Throughout the  article, there are lot of typological errors. Please rectify them:

Some are: Line 186: Check the spelling of "phylogenetic".

Line 214: Check the spelling of "displaying"

Line 218-221: The sentence is too lengthy that makes its meaning unclear. Please reframe it. 

Author Response

 #R1

The study by Leopardi et al. describes the circulation of canine parvoviruses and feline parvovirus in different animal species. The authors have also tried to evaluate how illegal import of animals can be source of different viral strains carrying novel antigens.

Dear reviewer, many thanks for the time spent to provide your valuable comments, which have all been addressed and can be found highlighted here below.

The study is indeed important, however, I have some queries: 

In the introduction section, the authors have given the details of structure, prevalence and classification of CPV and FPV. The authors have highlighted the illegal import of animals in the title but have not given its description in the introduction section. To make it more interesting to the readers and to highlight the consequences of illegal import, the authors should also include a paragraph highlighting the status, consequences and associated economic losses due to illegal import of animals. 

AR: the comment has been addressed accordingly and a new paragraph included.

In Results section:

The percent positivity of red foxes was found to be zero. However, this point has not been discussed much by the authors. Please include some more discussion by highlighting the possible reasons like lack of receptors etc.

AR: in the previous version of the manuscript we had discussed the negative result in foxes. Indeed, we did expect such a negative result, based on the total number of tested individuals and we discussed this result as follows: “Of note, all foxes tested negative in our study despite the number of individuals tested, thus excluding a prevalence of at least 15% considering the estimated fox population in Friuli Venezia Giulia. Although this result was expected based on previous finding estimating 2.8% prevalence of FPV in red foxes in other areas of Italy [13], enhanced surveil-lance in this species should be implemented in light of its susceptibility to both FPV and CPV viruses [14,26] and its well-known proximity with human settlements.” We have addressed your comment by further discussing the susceptibility of the red foxes to parvoviruses. Indeed, foxes are known to harbor a carnivore protoparvovirus species 4, which differently from carnivore protoparvovirus 1 is not detected by the real time PCR protocol applied in this study.

Some of the data in discussion has been explained up to a level that seems to be the repetition of results. Please get rid of that part. The authors should put more stress on discussion part rather than explaining the results.

AR: many thanks for this comment. We have addressed it throughout the text and we hope the revised version of the manuscript is now acceptable for publication.

Throughout the article, there are lot of typological errors. Please rectify them:

Some are: Line 186: Check the spelling of "phylogenetic".

Line 214: Check the spelling of "displaying"

AR: The typological errors have been accurately checked,

Line 218-221: The sentence is too lengthy that makes its meaning unclear. Please reframe it. 

AR: the sentence has been reframed accordingly.

Reviewer 2 Report

Overview and general recommendation:

In this study, Authors sequenced and analysed the CPV-2 and FPV strains from illegally imported domestic dogs and wildlife in north eastern Italy. This short but descriptive study is overall well written and descriptive, adding new data on the links for the viruses circulating in domestic and wild animals, as well as on the current viral spreading. Despite the potential interesting data, this study showed some limitations and needs some clarifications. I added some comments and suggestions to Authors, to improve the description throughout the manuscript with the aim to overcome these limits. I explained these comments and suggestions in more details below.

Major comments:

-          Lines 38-41: As expressed in the cited reference, “Canine parvovirus type-2 (CPV-2) arose in the mid-1970s as a variant of a virus similar to but distinct from FPV”: I suggest to revise according to this phrase. Moreover, I suggest to carefully check the use of the names of the CPV-2 type or variants (CPV-2, CPV-2a, CPV-2b, CPV-2c, new CPV-2a, new CPV-2b) throughout the whole text.

-          Lines 48-49: The active circulation of the original CPV-2 type is not restricted to South America (this evidence is not even reported in the cited references) but its evidence (mainly by molecular assays) is likely referred to the shedding of vaccinal CPV-2 strains, as correctly noticed in the second part of the sentence. I suggest to reconsider this phrase.

-          Lines 140-142: according to this phrase, I suggest to better clarify the number of sequenced strains in the M/M section (i.e., at line 90 where Authors stated “of target positive samples”) and, since 74 out of 290 positive dogs were sequenced, to include if any criteria (i.e. period of collection) was adopted for the selection of the positive samples for sequencing or if were randomly selected. 

-          - Lines 165-167: the topology of the phylogenetic trees showed that sequences of viral strains from wildlife are included in a separate branch, along with the sequences of other FPV strains, except for that one from the gray wolf, included in a cluster within the sequences of CPV-2 strains. The typing of the viral strains harboured from tested animals was deduced from other sequence analyses. I suggest to rephrase this part.    

-          Line 182: Authors referred to partial VP2 gene sequences or partial genome sequences? Both in M/M and results sections, Authors discussed the phylogenetic analyses based on the complete genome sequences and the complete VP2 gene sequences but only the phylogenetic tree based on the complete VP2 gene sequences was included in the main text, while the other one (based on the complete genome) was included as a supplementary file, despite it was discussed in the main text: according to the Authors guidelines, I suggest to include both phylogenetic trees in the main text to allow a clear reading of the study.

-          Lines 224-230: this CPV-2c strain is not related with strains generically found “across the world” but, as highlighted in the phylogenetic trees, with CPV-2c, so far defined, with Asian origins, reported also in Italy in domestic dogs and in a wolf in central Italy in 2020 (MT454909). Moreover, in the discussion section (lines 296-297) this strain was only correlated with sequences found in two imported puppies, and not with CPV-2c strains of asian origins reported in reference [9], similar to those already circulating in domestic dogs in southern Italy, rather than to the CPV-2b strains in Italy [22] and CPV-2b/-2c in Spain [10] from wolves (not so genetically close related). Following this comment, Authors could improve the related discussion.

-          Lines 276-277: Authors could provide a comment in this section concerning the evidence of the “old” CPV-2b variant in domestic dogs: is it related to other known vaccinal stains? Any other hypothesis?

-          Lines 339-340: since this study was based on a molecular approach, it is not possible to associate the severity of the clinical signs or, particularly, the mortality with the vaccinal CPV-2 strains. These evidences deserve other analitical approaches and probably, particularly for the negative outcome, are related to other causes.

Minor comments:

-          Title: I suggest to change “CPV” with “CPV-2”.

-          Line 38: use italics characters for the family name.

-          Line 42: I suggest to replace “classification” with “typing” and “canine parvoviruses” with “CPV-2”.

-          Line 44: I suggest to add “(at aa residues 297 and 426)” after “substitutions”

-          Lines 47-48: I suggest to change “antigenic types” with “antigenic and genetic variants”, to replace the variants names with “CPV-2a, -2b, -2c”, to consider that the CPV-2c variant arose more recently than the cited new CPV-2a/-2b (referred as “the most recent subtypes”), to replace “new 2° and new 2b” with “new CPV-2a and new CPV-2b”, and to add a comma before “that”.

-          Line 69: “there are” and “in” (in brackets) could be removed. 

-          Lines 86-87: lowercase letters should be used for canine parvovirus and feline panleukopenia virus; use the singular for feline panleukopenia virus. Moroever, Authors cited a molecular assay to test both CPV-2 and FPV, despite the cited reference reports “targeting a region of the VP2 gene of CPV-2 developed at the IZSVe”: does this method allow to test both CPV-2 and FPV?

-          Lines 90-91: somewhat conflict in this phrase: the complete genome “of target positive samples” was characterized but “selected” strains” were included in the analysis? Moreover, at least in this phrase and at line 98, I suggest to include “near” before “complete”.

-          Table 1: what does “original primers” mean? Please change “Ref” with “Reference”; Authors can consider to the replace “Perez et. al 2014” with “[12]”.

-          Line 102: please include details of the producer.

-          Lines 127-128: “paying attention…our study” is not referred to “positive original samples” at the beginning of the sentence?

-          Line 133: I suggest to replace with “the typing of the CPV-2/FPV strains”

-          Line 134: I suggest to replace with “considered VP2 amino acid residues at positions..”

-          Line 137: I suggest to remove “among”

-          Line 138: I suggest to include the percentage of total tested positive samples in brackets, close to “n=298”.

-          Line 140: replace “turned out” with “tested”.

-          Lines 142-143: “majority of CPV-2..characterized” deserves a review. Moreover, I suggest to revise (both here and in the related table) the names of the CPV-2 type and variants, according to the first major comment.

-          Lines 143-145: I suggest to revise as follow: “Whereas, viral strains from wildlife were characterized as FPV (from Eurasian bdger and golden jackal) and CPV-2c (from the gray wolf) (Table 3).”.

-          Table 3: I suggest to replace the title of the fourth column with “Viral type/variant (% out of positive samples)”.

-          Lines 148-154: consider to move this part at line 140.

-          Line 155: “succesfully” conflicts with limitations reported at lines 158-161.

-          Lines 165-166: for “the two approches” Authors intended the phylogenetic analyses based on the whole genome sequences and on the VP2 gene sequences?

-          Figure 1: “of Carnivore protoparvovirus 1”; “on the complete VP2 gene sequences”;

-          Line 182: published available sequences detected in Italian dogs not included “only” partial genomes (almost all included only partial genomes), as well as for FPV sequences: I suggest to consider “Almost all published available CPV-2/FPV sequences detected in the italian domestic dogs and cats, as well as in European wildlife accounted for partial genomes,..”.

-          Lines 182-183: “to a certain extent” could be removed.

-          Line 190: “while…genome.” could be removed since appears as a repetition of the same concept.

-          Line 190: For clarity, I suggest shifting this sentence (FPV sequences generated..) to the next line.

-          Line 191: I suggest to add (Figure 1) after “other” and to replace “,” with “:”.

-          Lines 191-192: I suggest to change with “with the FPV sequence (OP595740) from Eurasian badger characterised in this study and FPV sequences from italian cats and wildlife, another one (OP588006) with FPV sequence from Canadian wildlife, and the last one (OP587998) wih FPV sequences from italian domestic cats. As for the FPV strain from Eurasian badgers, … genomes (Figure S1).”

-          Lines 201-202: Revise the names of the CPV-2 variants.

-          Lines 202-203: I suggest to consider to revise this sentence as follow: “The high prevalence of the new CPV-2a variant in illegally imported dogs is similar to those recently observed in dogs in the same italian geographical area [6].”. 

-          Lines 204-206: for more clarity, consider if this part could be moved at line 200.

-          Lines 216-221: similarly, consideri if this part could be moved at line 203.

-          Line 207: replace “eighties” with “80s”.

-          Lines 210-212: the exact meaning of “CPV-2INT”, the name of the CPV strain included in this commercial vaccine (I suggest to include the CPV strain rather than the commercial name and, if available, the accession number), as well as the meaning of the reference [17] here reported, are not clear.

-          Line 215: since the screening for other pathogens was not included in this manuscript, I suggest to include, at least, “(data not shown)” at the end of the sentence.

-          Line 225: CPV-2c rather than 2c

-          Line 235: I suggest to replace “As” with “Despite”.

-          Line 237: The high prevalence of CPV-2 in tested dogs highlights the risk posed by the illegal introduction. Authors could also included a similar comment in the discussion.

-          Line 244: replace “eighties” with “80s”.

-          Line 252: what does “even if with sporadic exceptions” mean?

-          Line 255: as previously noticed, I suggest to include the name of the vaccinal CPV-2 strain, rather than the commercial name (this name is referred to the old name of the company, no more available in the market with the same name). Moreover, since puppies were probably vaccinated before the introduction in Italy, the evidence of the original CPV-2 type is not surprising.

-          Line 273: Revise the names of the CPV-2 variants.

-          Line 277: Revise the names of the new CPV-2b variants.

-          Line 281: check “CPV2-b”

-          Line 300: check if “southern” should be removed.

-          Line 311: I suggest to add somewhat similar to “and the evidence for FPV strains related to viruses to date reported only in Canada”

-          Line 335: check “CPV-new2a”

-          Line 339: check if “central Italy”.

-          Reference [2]: check the title.

-          Figure 1 and Supplementary Figure 1: country associated to the sequence MF069446 is missing.

Author Response

Overview and general recommendation:

In this study, Authors sequenced and analysed the CPV-2 and FPV strains from illegally imported domestic dogs and wildlife in north eastern Italy. This short but descriptive study is overall well written and descriptive, adding new data on the links for the viruses circulating in domestic and wild animals, as well as on the current viral spreading. Despite the potential interesting data, this study showed some limitations and needs some clarifications. I added some comments and suggestions to Authors, to improve the description throughout the manuscript with the aim to overcome these limits. I explained these comments and suggestions in more details below.

AR: Dear reviewer, by amending the manuscript according to your constructive comments and advice throughout the manuscript, we hope we have now provided a substantial improvement to the study. 

Major comments:

-          Lines 38-41: As expressed in the cited reference, “Canine parvovirus type-2 (CPV-2) arose in the mid-1970s as a variant of a virus similar to but distinct from FPV”: I suggest to revise according to this phrase. Moreover, I suggest to carefully check the use of the names of the CPV-2 type or variants (CPV-2, CPV-2a, CPV-2b, CPV-2c, new CPV-2a, new CPV-2b) throughout the whole text.

AR: the sentence has been rephrased and the use of the correct names has been checked throughout the whole text accordingly.

-          Lines 48-49: The active circulation of the original CPV-2 type is not restricted to South America (this evidence is not even reported in the cited references) but its evidence (mainly by molecular assays) is likely referred to the shedding of vaccinal CPV-2 strains, as correctly noticed in the second part of the sentence. I suggest to reconsider this phrase.

AR: the sentence has been reconsidered and addressed accordingly.

-          Lines 140-142: according to this phrase, I suggest to better clarify the number of sequenced strains in the M/M section (i.e., at line 90 where Authors stated “of target positive samples”) and, since 74 out of 290 positive dogs were sequenced, to include if any criteria (i.e. period of collection) was adopted for the selection of the positive samples for sequencing or if were randomly selected. 

AR: the selection criteria have been better clarified in the M&M section. Indeed, we stratified the sampling according to the year of finding (2018-2021) and the type of sample (feces Vs intestines).

-          - Lines 165-167: the topology of the phylogenetic trees showed that sequences of viral strains from wildlife are included in a separate branch, along with the sequences of other FPV strains, except for that one from the gray wolf, included in a cluster within the sequences of CPV-2 strains. The typing of the viral strains harboured from tested animals was deduced from other sequence analyses. I suggest to rephrase this part.    

AR: the sentence has now been rephrased accordingly.

-          Line 182: Authors referred to partial VP2 gene sequences or partial genome sequences? Both in M/M and results sections, Authors discussed the phylogenetic analyses based on the complete genome sequences and the complete VP2 gene sequences but only the phylogenetic tree based on the complete VP2 gene sequences was included in the main text, while the other one (based on the complete genome) was included as a supplementary file, despite it was discussed in the main text: according to the Authors guidelines, I suggest to include both phylogenetic trees in the main text to allow a clear reading of the study.

AR: The phylogenetic tree of full genome sequences has been included as Figure 2 and the  text changed accordingly.

-          Lines 224-230: this CPV-2c strain is not related with strains generically found “across the world” but, as highlighted in the phylogenetic trees, with CPV-2c, so far defined, with Asian origins, reported also in Italy in domestic dogs and in a wolf in central Italy in 2020 (MT454909). Moreover, in the discussion section (lines 296-297) this strain was only correlated with sequences found in two imported puppies, and not with CPV-2c strains of asian origins reported in reference [9], similar to those already circulating in domestic dogs in southern Italy, rather than to the CPV-2b strains in Italy [22] and CPV-2b/-2c in Spain [10] from wolves (not so genetically close related). Following this comment, Authors could improve the related discussion.

AR: the authors wish to thank the reviewer for having noticed this point. Indeed, we checked the aa residues retained by the wolf CPV-2 sequence and compared them to those found in the wolf in Southern Italy and in the imported puppies. This comparison allowed us to better discuss the possibility that this virus might circulate independently in the wolf population, although the occurrence of spillover events from the domestic dogs still remains. We have therefore addressed the comment and included this point in the discussion section.

-          Lines 276-277: Authors could provide a comment in this section concerning the evidence of the “old” CPV-2b variant in domestic dogs: is it related to other known vaccinal stains? Any other hypothesis?

AR: In 2008, CPV-2b reappeared in Italy after ten years, and since then has maintained a low prevalence, raising several doubts on the origin of such a renewed circulation. Indeed, CPV-2b has been included as a vaccine component, due to its notable capability of inducing a broad-spectrum immunity against both CPV-2a and CPV-2c strains. The origin of the CPV-2b sequences identified in the illegally imported puppies in the present study could therefore be due to vaccine strain occurrence rather than to an active circulation of the “old” CPV-2b variant. Of note, all dogs carrying CPV-2b sequences displayed no signs of acute gastroenteritis. This comment has been included along the text in the Discussion session. The authors acknowledge that this point was not fully raised throughout the manuscript.

-          Lines 339-340: since this study was based on a molecular approach, it is not possible to associate the severity of the clinical signs or, particularly, the mortality with the vaccinal CPV-2 strains. These evidences deserve other analitical approaches and probably, particularly for the negative outcome, are related to other causes.

AR: the authors acknowledge that the absence of histopathological findings (i.e. immunohistochemistry) was the main weakness of the study. The sentence has been changed accordingly.

Minor comments:

-          Title: I suggest to change “CPV” with “CPV-2”.

AR: addressed accordingly.

-          Line 38: use italics characters for the family name.

AR: addressed accordingly.

-          Line 42: I suggest to replace “classification” with “typing” and “canine parvoviruses” with “CPV-2”.

AR: addressed accordingly.

-          Line 44: I suggest to add “(at aa residues 297 and 426)” after “substitutions”

AR: addressed accordingly.

-          Lines 47-48: I suggest to change “antigenic types” with “antigenic and genetic variants”, to replace the variants names with “CPV-2a, -2b, -2c”, to consider that the CPV-2c variant arose more recently than the cited new CPV-2a/-2b (referred as “the most recent subtypes”), to replace “new 2° and new 2b” with “new CPV-2a and new CPV-2b”, and to add a comma before “that”.

AR: addressed accordingly.

-          Line 69: “there are” and “in” (in brackets) could be removed

AR: addressed accordingly.

-          Lines 86-87: lowercase letters should be used for canine parvovirus and feline panleukopenia virus; use the singular for feline panleukopenia virus. Moroever, Authors cited a molecular assay to test both CPV-2 and FPV, despite the cited reference reports “targeting a region of the VP2 gene of CPV-2 developed at the IZSVe”: does this method allow to test both CPV-2 and FPV?

AR: addressed accordingly. The method was developed in a conserved portion of the VP2 region and is currently used for the molecular diagnosis of both CPV-2 and FPV.

-          Lines 90-91: somewhat conflict in this phrase: the complete genome “of target positive samples” was characterized but “selected” strains” were included in the analysis? Moreover, at least in this phrase and at line 98, I suggest to include “near” before “complete”.

AR: addressed accordingly.

-          Table 1: what does “original primers” mean? Please change “Ref” with “Reference”; Authors can consider to the replace “Perez et. al 2014” with “[12]”.

AR: the authors acknowledge that references in Table 1 needed to be improved and have amended according to the suggestions.

-          Line 102: please include details of the producer.

AR: addressed accordingly.

-          Lines 127-128: “paying attention…our study” is not referred to “positive original samples” at the beginning of the sentence?

AR: Datasets included positive original samples and the first three non-identical best matches for each sequence, as determined using BLAST reference strains previously associated either with dogs in the study area [6] or with wildlife across the world. The selection from the public database included sequences from Italy or wild species tested positive in our study. The authors acknowledge that the passage was unclear and rephrased it to better describe this issue.  

-          Line 133: I suggest to replace with “the typing of the CPV-2/FPV strains”

AR: replaced accordingly.

-          Line 134: I suggest to replace with “considered VP2 amino acid residues at positions..”

AR: replaced accordingly.

-          Line 137: I suggest to remove “among”

AR: removed accordingly.

-          Line 138: I suggest to include the percentage of total tested positive samples in brackets, close to “n=298”.

AR: addressed accordingly.

-          Line 140: replace “turned out” with “tested”.

AR: replaced accordingly.

-          Lines 142-143: “majority of CPV-2..characterized” deserves a review. Moreover, I suggest to revise (both here and in the related table) the names of the CPV-2 type and variants, according to the first major comment.

AR: addressed according to the first major comment.

-          Lines 143-145: I suggest to revise as follow: “Whereas, viral strains from wildlife were characterized as FPV (from Eurasian bdger and golden jackal) and CPV-2c (from the gray wolf) (Table 3).”.

AR: revised accordingly.

-          Table 3: I suggest to replace the title of the fourth column with “Viral type/variant (% out of positive samples)”.

AR: revised accordingly.

-          Lines 148-154: consider to move this part at line 140.

AR: moved accordingly.

-          Line 155: “succesfully” conflicts with limitations reported at lines 158-161.

AR: addressed accordingly.

-          Lines 165-166: for “the two approches” Authors intended the phylogenetic analyses based on the whole genome sequences and on the VP2 gene sequences?

AR: the sentence has been rephrased accordingly.

-          Figure 1: “of Carnivore protoparvovirus 1”; “on the complete VP2 gene sequences”;

AR: addressed accordingly.

-          Line 182: published available sequences detected in Italian dogs not included “only” partial genomes (almost all included only partial genomes), as well as for FPV sequences: I suggest to consider “Almost all published available CPV-2/FPV sequences detected in the italian domestic dogs and cats, as well as in European wildlife accounted for partial genomes,..”.

  1. The sentence has now been rephrased.

-          Lines 182-183: “to a certain extent” could be removed.

  1. Removed accordingly.

-          Line 190: “while…genome.” could be removed since appears as a repetition of the same concept.

AR: removed accordingly.

-          Line 190: For clarity, I suggest shifting this sentence (FPV sequences generated..) to the next line.

AR: shifted accordingly.

-          Line 191: I suggest to add (Figure 1) after “other” and to replace “,” with “:”.

AR: addressed accordingly.

-          Lines 191-192: I suggest to change with “with the FPV sequence (OP595740) from Eurasian badger characterised in this study and FPV sequences from italian cats and wildlife, another one (OP588006) with FPV sequence from Canadian wildlife, and the last one (OP587998) wih FPV sequences from italian domestic cats. As for the FPV strain from Eurasian badgers, … genomes (Figure S1).”

AR: the sentences have now been changed according to the reviewer’s inputs.

-          Lines 201-202: Revise the names of the CPV-2 variants.

AR: the names have been revised accordingly.

-          Lines 202-203: I suggest to consider to revise this sentence as follow: “The high prevalence of the new CPV-2a variant in illegally imported dogs is similar to those recently observed in dogs in the same italian geographical area [6].”. 

AR: changed accordingly.

-          Lines 204-206: for more clarity, consider if this part could be moved at line 200.

AR: moved accordingly.

-          Lines 216-221: similarly, consideri if this part could be moved at line 203.

AR: these sentences have been not moved according to the reviewer’ comment, as they refer to the description of a CPV-2 variant. The authors retained the original position along the text.

-          Line 207: replace “eighties” with “80s”.

AR: replaced accordingly.

-          Lines 210-212: the exact meaning of “CPV-2INT”, the name of the CPV strain included in this commercial vaccine (I suggest to include the CPV strain rather than the commercial name and, if available, the accession number), as well as the meaning of the reference [17] here reported, are not clear.

AR: the authors agreed with your previous comments and decided to delete any reference to the Original or Intervet strain. Sequences not classifiable as CPV-2 variants are therefore referred to as CPV-2. Nevertheless, Calatayud and co-authors first demonstrated the aa signature of the attenuated strains included in two commercially available vaccine formulations, and we therefore decided to include this reference for sake of clarity. The GB accession number of the sequence referred by Catalyud and co-author has been mentioned along the text.

-          Line 215: since the screening for other pathogens was not included in this manuscript, I suggest to include, at least, “(data not shown)” at the end of the sentence.

AR: Included accordingly.

-          Line 225: CPV-2c rather than 2c

AR: addressed accordingly.

-          Line 235: I suggest to replace “As” with “Despite”.

AR: changed accordingly.

-          Line 237: The high prevalence of CPV-2 in tested dogs highlights the risk posed by the illegal introduction. Authors could also included a similar comment in the discussion.

AR: addressed accordingly.

-          Line 244: replace “eighties” with “80s”.

AR: moved accordingly.

-          Line 252: what does “even if with sporadic exceptions” mean?

AR: the authors were referring to some CPV-2 circulation reported in literature. However, based on the reviewer’ s comment, this aside has been removed from the sentence in order to allow more clarity for the reader.

-          Line 255: as previously noticed, I suggest to include the name of the vaccinal CPV-2 strain, rather than the commercial name (this name is referred to the old name of the company, no more available in the market with the same name). Moreover, since puppies were probably vaccinated before the introduction in Italy, the evidence of the original CPV-2 type is not surprising.

AR: the name of the vaccine has been removed from the manuscript. The authors wish to thank the reviewer for this constructive comment.

-          Line 273: Revise the names of the CPV-2 variants.

AR: revised accordingly.

-          Line 277: Revise the names of the new CPV-2b variants.

AR: revised accordingly.

-          Line 281: check “CPV2-b”

AR: one part has been included to discuss about the presence of the CPV-2b variant. The authors wish to thank the reviewer for this key comment.

-          Line 300: check if “southern” should be removed.

AR: the authors have checked the references and confirm that CPV-2 sequences have been detected in wolves in Central (Lazio) and Southern (Abruzzi, Molise and Calabria) Italian regions. Based on the information available from the literature, the sentence has remained unchanged.

-          Line 311: I suggest to add somewhat similar to “and the evidence for FPV strains related to viruses to date reported only in Canada”

AR: the authours have thoroughly tried to find the lines the reviewers' suggestions refer to and were unable to address this comment. Indeed, the reviewer suggests including a comment related to North American wildlife ecopathology, whilst the study focuses on wildlife species not present in the Canadian territory.

-          Line 335: check “CPV-new2a”

AR: addressed accordingly.

-          Line 339: check if “central Italy”.

AR: In Italian wolves, the CPV-2c related to the sequence found in the present study (OP595742) originated from the Abruzzi region (MT450909), which  is currently considered Southern Italy by national authorities. Based on this assumption, the sentence has remained unchanged.

-          Reference [2]: check the title.

AR: addressed accordingly.

-          Figure 1 and Supplementary Figure 1: country associated to the sequence MF069446 is missing.

AR: addressed accordingly.

Round 2

Reviewer 2 Report

1. Overview and general recommendation:

I reviewed this manuscript and I found it improved in almost all parts. Despite Authors have satisfactorily addressed almost all comments and suggestions, few edits are still necessary. I explained these suggestions in more details below.

Minor comments:

- Line 45: please, change “b” with “CPV-2b”

- Line 59: I suggest to change “strain” with “type”.

- Lines 78-79: I suggest to use italics characters for Salmonella and Microsporum canis.

- Lines 130-131: Please, consider to rephrase this part as follows: “...genome of all wildlife strains

and of strains from selected samples of seventy-four positive dogs.”

- Table 1: as in the previous review stage, I suggest to change “Ref” with “Reference”

- Line 196: I suggest to replace “are often” with “were”.

- Line 201: Please replace “2b, and 2c” with “CPV-2b, and CPV-2c”

- Table 3: In the second row of the “Viral type/variant” column, I suggest to replace directly with

“CPV-2 (12.2), new CPV-2a (74.3), CPV-2b (8.1), new CPV-2b (2.7), CPV-2c (2.7)”

- Line 305: please, replace “new2b” with “new CPV-2b”.

- Lines 317, 319, and 320: please, replace “CPV new-2a” with “new CPV-2a”

- Line 501: move “species” before the comma

Author Response

Dear reviewer, many thanks for your prompt revision o the manuscript. We have addressed all remarks as explained below.

Minor comments:

- Line 45: please, change “b” with “CPV-2b”

AR: changed accordingly.

- Line 59: I suggest to change “strain” with “type”.

AR: changed accordingly.

- Lines 78-79: I suggest to use italics characters for Salmonella and Microsporum canis.

AR: addressed accordingly.

- Lines 130-131: Please, consider to rephrase this part as follows: “...genome of all wildlife strains

and of strains from selected samples of seventy-four positive dogs.”

AR: addressed accordingly.

- Table 1: as in the previous review stage, I suggest to change “Ref” with “Reference”

AR: addressed accordingly.

- Line 196: I suggest to replace “are often” with “were”.

AR: replaced accordingly.

- Line 201: Please replace “2b, and 2c” with “CPV-2b, and CPV-2c”

AR: replaced accordingly.

- Table 3: In the second row of the “Viral type/variant” column, I suggest to replace directly with

“CPV-2 (12.2), new CPV-2a (74.3), CPV-2b (8.1), new CPV-2b (2.7), CPV-2c (2.7)”

AR: addressed accordingly.

- Line 305: please, replace “new2b” with “new CPV-2b”.

AR: addressed accordingly.

- Lines 317, 319, and 320: please, replace “CPV new-2a” with “new CPV-2a”

AR: addressed accordingly.

- Line 501: move “species” before the comma

AR: moved accordingly